# Augmented reality powers a cognitive assistant for the blind

Yang Liu[1,2], Noelle RB Stiles[1,3], Markus Meister[1]*

[1]Division of Biology and Biological Engineering, California Institute of Technology, Pasadena, United States; [2]Computation and Neural Systems Program, California Institute of Technology, Pasadena, United States; [3]Institute for Biomedical Therapeutics, Keck School of Medicine, University of Southern California, Los Angeles, United States

**Abstract** To restore vision for the blind, several prosthetic approaches have been explored that convey raw images to the brain. So far, these schemes all suffer from a lack of bandwidth. An alternate approach would restore vision at the cognitive level, bypassing the need to convey sensory data. A wearable computer captures video and other data, extracts important scene knowledge, and conveys that to the user in compact form. Here, we implement an intuitive user interface for such a device using augmented reality: each object in the environment has a voice and communicates with the user on command. With minimal training, this system supports many aspects of visual cognition: obstacle avoidance, scene understanding, formation and recall of spatial memories, navigation. Blind subjects can traverse an unfamiliar multi-story building on their first attempt. To spur further development in this domain, we developed an open-source environment for standardized benchmarking of visual assistive devices.
DOI: https://doi.org/10.7554/eLife.37841.001

## Introduction

About 36 million people are blind worldwide (*Bourne et al., 2017*). In industrialized nations, the dominant causes of blindness are age-related diseases of the eye, all of which disrupt the normal flow of visual data from the eye to the brain. In some of these cases, biological repair is a potential option, and various treatments are being explored involving gene therapy, stem cells, or transplantation (*Scholl et al., 2016*). However, the dominant strategy for restoring vision has been to bring the image into the brain through alternate means. The most direct route is electrical stimulation of surviving cells in the retina (*Stingl and Zrenner, 2013*; *Weiland and Humayun, 2014*) or of neurons in the visual cortex (*Dobelle et al., 1974*). Another option involves translating the raw visual image into a different sensory modality (*Loomis et al., 2012*; *Maidenbaum et al., 2014*; *Proulx et al., 2016*), such as touch (*Stronks et al., 2016*) or hearing (*Auvray et al., 2007*; *Capelle et al., 1998*; *Meijer, 1992*). So far, none of these approaches has enabled any practical recovery of the functions formerly supported by vision. Despite decades of efforts all users of such devices remain legally blind (*Luo and da Cruz, 2016*; *Stingl et al., 2017*; *Striem-Amit et al., 2012*; *Stronks et al., 2016*). While one can certainly hope for progress in these domains, it is worth asking what are the fundamental obstacles to restoration of visual function.

The human eye takes in about 1 gigabit of raw image information every second, whereas our visual system extracts from this just tens of bits to guide our thoughts and actions (*Pitkow and Meister, 2014*). All the above approaches seek to transmit the raw image into the brain. This requires inordinately high data rates. Further, the signal must arrive in the brain in a format that can be interpreted usefully by the visual system or some substitute brain area to perform the key steps of knowledge acquisition, like scene recognition and object identification. None of the technologies available

*For correspondence:
meister@caltech.edu

today deliver the high data rate required to retain the relevant details of a scene, nor do they produce a neural code for the image that matches the expectations of the human brain, even given the prodigious degree of adaptive plasticity in the nervous system.

Three decades ago, one of the pioneers of sensory substitution articulated his vision of a future visual prosthesis (*Collins, 1985*): "I strongly believe that we should take a more sophisticated approach, utilizing the power of artificial intelligence for processing large amounts of detailed visual information in order to substitute for the missing functions of the eye and much of the visual pre-processing performed by the brain. We should off-load the blind travelers' brain of these otherwise slow and arduous tasks which are normally performed effortlessly by the sighted visual system". Whereas at that time the goal was hopelessly out of reach, today's capabilities in computer vision, artificial intelligence, and miniaturized computing power are converging to make it realistic.

Here, we present such an approach that bypasses the need to convey the sensory data entirely, and focuses instead on the important high-level knowledge, presented at a comfortable data rate and in an intuitive format. We call the system CARA: a cognitive *a*ugmented *r*eality *a*ssistant for the blind.

## Results

### Design principles

CARA uses a wearable augmented reality device to give voices to all the relevant objects in the environment (*Figure 1A*). Unlike most efforts at scene sonification (*Bujacz and Strumiłło, 2016*; *Csapó and Wersényi, 2013*), our system communicates through natural language. Each object in the scene can talk to the user with a voice that comes from the object's location. The voice's pitch increases as the object gets closer. The user actively selects which objects speak through several modes of control (*Figure 1—figure supplement 1*): In Scan mode, the objects call out their names in sequence from left to right, offering a quick overview of the scene. In Spotlight mode, the object directly in front speaks, and the user can explore the scene by moving the head. In Target mode, the user selects one object that calls repeatedly at the press of a clicker. In addition, any surface in the space emits a hissing sound as a collision warning when the user gets too close (*Figure 1—figure supplement 1*).

The system is implemented on the Microsoft HoloLens (*Figure 1A*), a powerful head-mounted computer designed for augmented reality (*Hoffman, 2016*). The HoloLens scans all surfaces in the environment using video and infrared sensors, creates a 3D map of the surrounding space, and localizes itself within that volume to a precision of a few centimeters (*Figure 1—figure supplement 2*). It includes a see-through display for digital imagery superposed on the real visual scene; open ear speakers that augment auditory reality while maintaining regular hearing; and an operating system that implements all the localization functions and provides access to the various sensor streams.

Any cognitive assistant must both acquire knowledge about the environment and then communicate that knowledge to the user. Tracking and identifying objects and people in a dynamic scene still presents a challenge, but those capabilities are improving at a remarkable rate (*Jafri et al., 2014*; *Verschae and Ruiz-del-Solar, 2015*), propelled primarily by interests in autonomous vehicles (see also *Technical extensions* below). Anticipating that the acquisition problems will be solved shortly, we focus here on the second task, the interface to the user. Thus, we populated the real-space volume scanned by the HoloLens with virtual objects that interact with the user. The applications were designed using the Unity game development platform which allows tracking of the user's head in the experimental space; the simulation of virtual objects; the generation of speech and sounds that appear to emanate from specific locations; and interaction with the user via voice commands and a clicker.

### Human subject tests

After a preliminary exploration of these methods, we settled on a fixed experimental protocol and recruited seven blind subjects (*Figure 1D*). Subjects heard a short explanation of what to expect, then donned the HoloLens and launched into a series of four fully automated tasks without experimenter involvement. No training sessions were provided, and all the data were gathered within a 2 hr visit.

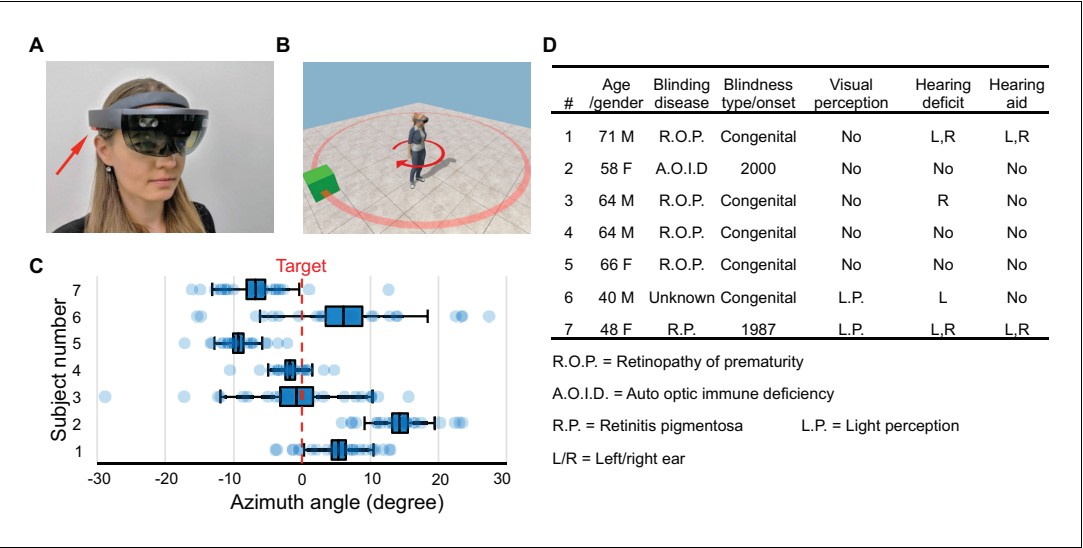

**Figure 1.** Hardware platform and object localization task. (**A**) The Microsoft HoloLens wearable augmented reality device. Arrow points to one of its stereo speakers. (**B**) In each trial of the object localization task, the target (green box) is randomly placed on a circle (red). The subject localizes and turns to aim at the target. (**C**) Object localization relative to the true azimuth angle (dashed line). Box denotes s.e.m., whiskers s.d. (**D**) Characteristics of the seven blind subjects.

DOI: https://doi.org/10.7554/eLife.37841.002

The following figure supplements are available for figure 1:

**Figure supplement 1.** Obstacle avoidance utility and active scene exploration modes.

DOI: https://doi.org/10.7554/eLife.37841.003

**Figure supplement 2.** Process of scene sonification.

DOI: https://doi.org/10.7554/eLife.37841.004

## Object localization

Here, we tested the user's ability to localize an augmented reality sound source (*Figure 1B*). A virtual object placed randomly at a 2 m distance from the subject called out 'box' whenever the subject pressed a clicker. The subject was asked to orient the head towards the object and then confirm the final choice of direction with a voice command. All subjects found this a reasonable request and oriented surprisingly well, with an accuracy of 3–12 degrees (standard deviation across trials, *Figure 1C*). Several subjects had a systematic pointing bias to one or the other side of the target (−9 to +13 deg, *Figure 1C*), but no attempt was made to correct for this bias. These results show that users can accurately localize the virtual voices generated by HoloLens, even though the software used a generic head-related transfer function without customization.

## Spatial memory

Do object voices help in forming a mental image of the scene (*Lacey, 2013*) that can be recalled for subsequent decisions? A panel of five virtual objects was placed in the horizontal plane 2 m from the subject, spaced 30 degrees apart in azimuth (*Figure 2A*). The subject scanned this scene actively using the Spotlight mode for 60 s. Then the object voices were turned off and we asked the subject to orient towards the remembered location of each object, queried in random order. All subjects performed remarkably well, correctly recalling the arrangement of all objects (*Figure 2B*, *Figure 2— figure supplement 1*) with just one error (1/28 trials). Even the overall scale of the scene and the absolute positions of the objects were reproduced well from memory, to an average accuracy of ~15 deg (rms deviation from true position, *Figure 2C–D*). In a second round, we shuffled the object positions and repeated the task. Here three of the subjects made a mistake, presumably owing to interference with the memory formed on the previous round. Sighted subjects who inspected the scene visually performed similarly on the recall task (*Figure 2—figure supplement 1*). These experiments suggest that active exploration of object voices builds an effective mental representation of the scene that supports subsequent recall and orientation in the environment.

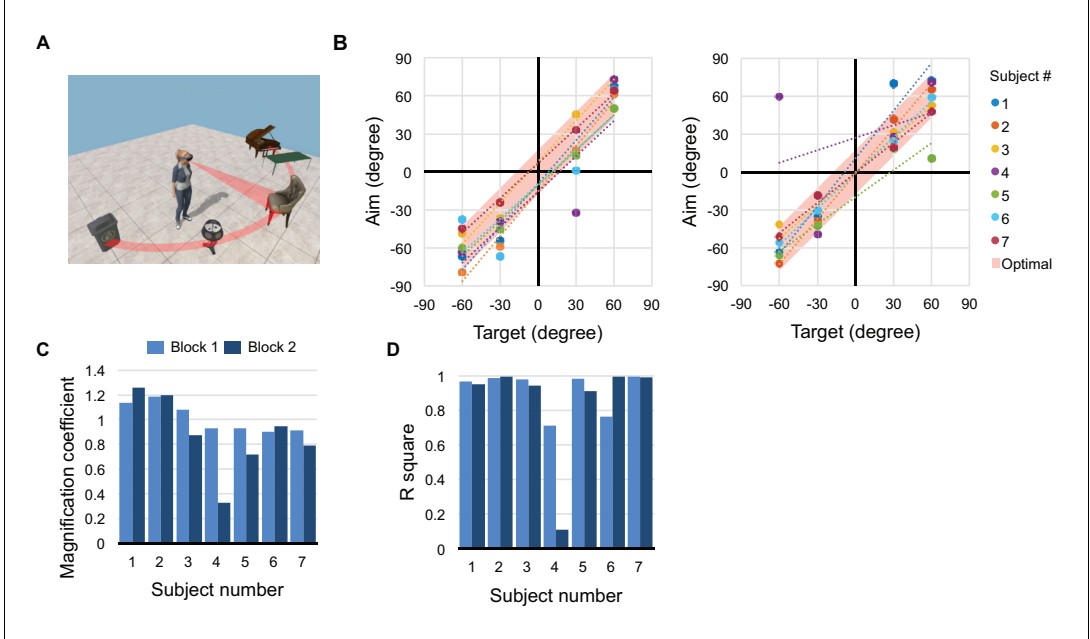

**Figure 2.** Spatial memory task. (**A**) Five objects are arranged on a half-circle; the subject explores the scene, then reports the recalled object identities and locations. (**B**) Recall performance during blocks 1 (left) and 2 (right). Recalled target angle potted against true angle. Shaded bar along the diagonal shows the 30 deg width of each object; data points within the bar indicate perfect recall. Dotted lines are linear regressions. (**C**) Slope and (**D**) correlation coefficient for the regressions in panel (**B**).

DOI: https://doi.org/10.7554/eLife.37841.005

The following figure supplement is available for figure 2:

**Figure supplement 1.** Mental imagery task supplementary data.

DOI: https://doi.org/10.7554/eLife.37841.006

## Direct navigation

Here, the subject was instructed to walk to a virtual chair, located 2 m away at a random location (*Figure 3A*). In Target mode, the chair called out its name on every clicker press. All subjects found the chair after walking essentially straight-line trajectories (*Figure 3B–C*, *Figure 3—figure supplement 1*). Most users followed a two-phase strategy: first localize the voice by turning in place, then walk swiftly toward it (*Figure 3—figure supplement 1D–E*). On rare occasions (~5 of 139 trials), a subject started walking in the opposite direction, then reversed course (*Figure 3—figure supplement 1C*), presumably owing to ambiguities in azimuthal sound cues (*McAnally and Martin, 2014*). Subject seven aimed consistently to the left of the target (just as in the task of *Figure 1*) and thus approached the chair in a spiral trajectory (*Figure 3C*). Regardless, for all subjects the average trajectory was only 11–25% longer than the straight-line distance (*Figure 3E*, *Figure 3—figure supplement 1A*).

For comparison, we asked subjects to find a real chair in the same space using only their usual walking aid (*Figure 3D*). These searches took on average eight times longer and covered 13 times the distance needed with CARA. In a related series of experiments we encumbered the path to the target with several virtual obstacles. Using the alarm sounds, our subjects weaved through the obstacles without collision (*Figure 3—figure supplement 2D*). Informal reports from the subjects confirmed that steering towards a voice is a natural function that can be performed automatically, leaving attentional bandwidth for other activities. For example, some subjects carried on a conversation while following CARA.

## Long-range guided navigation

If the target object begins to move as the subject follows its voice, it becomes a 'virtual guide'. We designed a guide that follows a precomputed path and repeatedly calls out 'follow me'. The guide monitors the subject's progress, and stays at most 1 m ahead of the subject. If the subject strays off

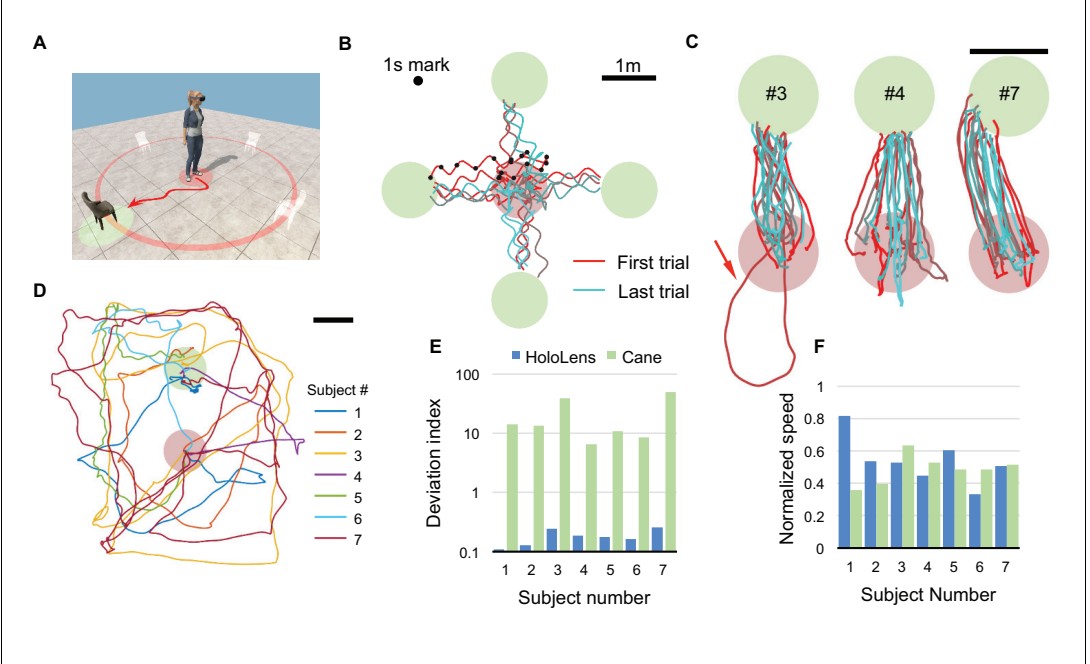

**Figure 3.** Direct navigation task. (**A**) For each trial, a target chair is randomly placed at one of four locations. The subject begins in the starting zone (red shaded circle), follows the voice of the chair, and navigates to the target zone (green shaded circle). (**B**) All raw trajectories from one subject (#6) including 1 s time markers. Oscillations from head movement are filtered out in subsequent analysis. (**C**) Filtered and aligned trajectories from all trials of 3 subjects (#3, 4, 7). Arrow highlights a trial where the subject started in the wrong direction. (**D**) Trajectories of subjects performing the task with only a cane and no HoloLens. (**E**) Deviation index, namely the excess length of the walking trajectory relative to the shortest distance between start and target. Note logarithmic axis and dramatic difference between HoloLens and Cane conditions. (**F**) Speed of each subject normalized to the free-walking speed.

DOI: https://doi.org/10.7554/eLife.37841.007

The following figure supplements are available for figure 3:

**Figure supplement 1.** Direct navigation task extended data.
DOI: https://doi.org/10.7554/eLife.37841.008
**Figure supplement 2.** Additional experimental functions.
DOI: https://doi.org/10.7554/eLife.37841.009

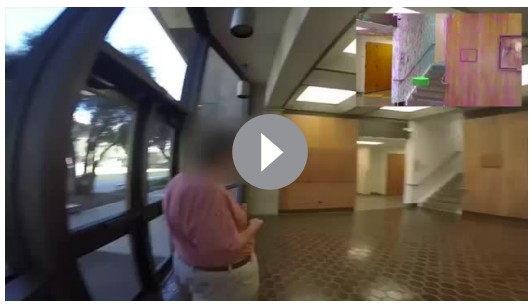

**Video 1.** Long-range navigation. A video recording of a subject navigating in the long-range navigation task. The top right panel shows the first person view of the subject recorded by the HoloLens.
DOI: https://doi.org/10.7554/eLife.37841.012

the path, the guide stops and waits for the subject to catch up. The guide also offers warnings about impending turns or a flight of stairs. To test this design, we asked subjects to navigate a campus building that had been pre-scanned by the Holo-Lens (*Figure 4A*, *Figure 4—figure supplement 1*). The path led from the ground-floor entrance across a lobby, up two flights of stairs, around several corners and along a straight corridor, then into a second floor office (*Figure 4B–C*). The subjects had no prior experience with this part of the building. They were told to follow the voice of the virtual guide, but given no assistance or coaching during the task.

All seven subjects completed the trajectory on the first attempt (*Figure 4B–C*, *Video 1*). Subject seven transiently walked off course (*Figure 4B*), due to her left-ward bias (*Figures 1C* and *3C*), then regained contact with the virtual guide. On a second attempt, this subject completed the task without straying. On average, this task required 119 s (range 73–159 s), a tolerable

**Figure 4.** Long-range guided navigation task. (**A**) 3D reconstruction of the experimental space with trajectories from all subjects overlaid. (**B and C**) 2D floor plans with all first trial trajectories overlaid. Trajectories are divided into three segments: lobby (Start – Start 2), stairwell (Start 2 – Start 3), and hallway (Start 3 – Destination). Red arrows indicate significant deviations from the planned path. (**D**) Deviation index (as in *Figure 3E*) for all segments by subject. Outlier corresponds to initial error by subject 7. Negative values indicate that the subject cut corners relative to the virtual guide. (**E**) Duration and (**F**) normalized speed of all the segments by subject.

DOI: https://doi.org/10.7554/eLife.37841.010

The following figure supplement is available for figure 4:

**Figure supplement 1.** Guided navigation trajectories.
DOI: https://doi.org/10.7554/eLife.37841.011

investment for finding an office in an unfamiliar building (*Figure 4E*). The median distance walked by the subjects was 36 m (*Figure 4D*), slightly shorter (~1%) than the path programmed for the virtual guide, because the subjects can cut corners (*Figure 4C*). The subjects' speed varied with difficulty along the route, but even on the stairs they proceeded at ~60% of their free-walking speed (*Figure 4F*). On arriving at the office, one subject remarked "That was fun! When can I get one?". Other comments from subjects regarding user experience with CARA are provided in 'Supplementary Observations'.

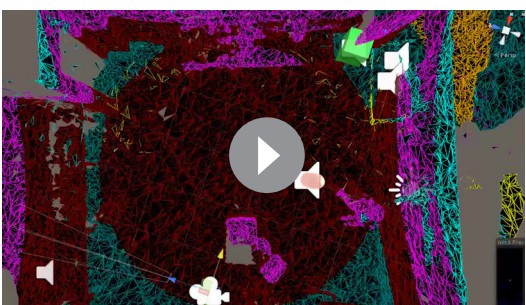

**Video 2.** Automatic wayfinding explained. A video demonstration of how automatic wayfinding works in a virtual environment.
DOI: https://doi.org/10.7554/eLife.37841.013

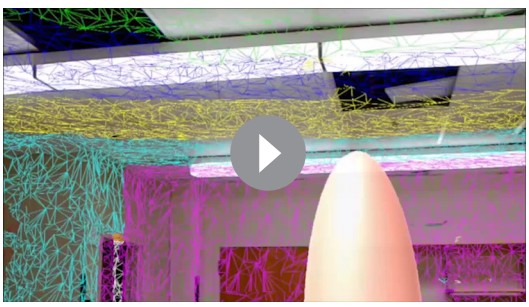

**Video 3.** Automatic wayfinding in an office. A point of view video demonstration of the automatic wayfinding function in an office space with obstacles. The path is calculated at the user's command based on the geometry of the office.
DOI: https://doi.org/10.7554/eLife.37841.014

## Technical extensions

As discussed above, the capabilities for identification of objects and people in a dynamic scene are rapidly developing. We have already implemented real-time object naming for items that are easily identified by the HoloLens, such as standardized signs and bar codes (*Sudol et al., 2010*) (*Figure 3—figure supplement 2A–B*). Furthermore, we have combined these object labels with a scan of the environment to compute in real time a navigable path around obstacles toward any desired target (*Figure 3—figure supplement 2C*, *Video 2*, *Video 3*). In the few months since our experimental series with blind subjects, algorithms have appeared that come close to a full solution. For example, YOLO (*Redmon and Farhadi, 2018*) will readily identify objects in a real time video feed that match one of 9000 categories. The algorithm already runs on the HoloLens and we are adopting it for use within CARA (*Figure 3—figure supplement 2F*).

## An open-source benchmarking environment for assistive devices

The dramatic advances in mobile computing and machine vision are enabling a flurry of new devices and apps that offer one or another assistive function for the vision impaired. To coordinate these developments one needs a reliable common standard by which to benchmark and compare different solutions. In several domains of engineering, the introduction of a standardized task with a quantitative performance metric has stimulated competition and rapid improvement of designs (*Berens et al., 2018*; *Russakovsky et al., 2015*).

On this background, we propose a method for the standardized evaluation of different assistive devices for the blind. The user is placed into a virtual environment implemented on the HTC Vive platform (*Wikipedia, 2018*). This virtual reality kit is widely used for gaming and relatively affordable. Using this platform, researchers anywhere in the world can replicate an identical environment and use it to benchmark their assistive methods. This avoids having to replicate and construct real physical spaces.

At test time the subject dons a wireless headset and moves freely within a physical space of 4 m x 4 m. The Vive system localizes position and orientation of the headset in that volume. Based on these data, the virtual reality software computes the subject's perspective of the virtual scene, and presents that view through the headset's stereo goggles. An assistive device of the experimenter's choice can use that same real-time view of the environment to guide a blind or blind-folded subject through the space. This approach is sufficiently general to accommodate designs ranging from raw sensory substitution – like vOICe (*Meijer, 1992*) and BrainPort (*Stronks et al., 2016*) – to cognitive assistants like CARA. The tracking data from the Vive system then serve to record the user's actions and evaluate the performance on any given task.

To illustrate this method, we constructed a virtual living room with furniture (*Figure 5A*). Within that space we defined three tasks that involve (1) scene understanding, (2) short-range navigation, and (3) finding a small object dropped on the floor. To enable blind subjects in these tasks we provided two assistive technologies: (a) the high-level assistant CARA, using the same principle of talking objects as described above on the HoloLens platform; (b) the low-level method vOICe that converts photographs to soundscapes at the raw image level (*Meijer, 1992*). The vOICe system was implemented using software provided by its inventor (*Seeing With Sound, 2018*).

Here, we report performance of four subjects, all normally sighted. Each subject was given a short explanation of both CARA and vOICe. The subject was allowed to practice (~10 min) with both methods by viewing the virtual scene while either CARA or vOICe provided translation to sound delivered by headphones. Then, the subjects were blindfolded and performed the three tasks with

sound alone. Each task consisted of 20 trials with randomly chosen goals, and a time limit of 60 s was applied to each trial.

On the first task, the subject stood in the middle of the virtual living room and was asked to locate one of the objects and point at it with the head. With CARA, subjects mostly used the Target mode to efficiently find the desired object, and located it based on the 3D sound cues, with a typical aiming error of 10 degrees (*Figure 5B*, bias of −0.2–13 deg, accuracy 4.5–27 deg). With vOICe, subjects reported great difficulty at identifying objects, despite the earlier opportunity to practice with visual feedback. Their aiming choices were statistically consistent with a uniform random distribution (*Figure 5B*).

On the second task, the subject was asked to walk from the middle of the arena toward a chair placed randomly in one of four locations, as in the directed navigation task of *Figure 3*. Using CARA, subjects found the chair efficiently, requiring only 10 s on average (*Figure 5C*). Using vOICe, most subjects meandered through the arena, on occasion encountering the chair by accident. Only one subject was able to steer toward the chair (*Figure 5—figure supplement 1*). None of the subjects were able to complete 20 trials within the 60 s time limit.

On the third task, the subject was asked to find a key that had fallen on the floor of the arena. To complete the task, the subject's head had to point toward the key at <1 m distance. Under those conditions, one can readily reach out and grasp the object. Using CARA, subjects found the key efficiently (*Figure 5D*). Using vOICe none of the subjects were able to locate the key (*Figure 5D*), although two of them encountered it once by accident (*Figure 5—figure supplement 1*).

These experiments illustrate the use of a standardized testing environment. Naive subjects performed well on these real-world tasks using CARA, but not using vOICe. It should be said that interpreting the vOICe sounds is very non-intuitive. Our subjects received the basic instructions offered on the vOICe web site and the 'Exercise' mode of vOICe (*Seeing With Sound, 2018*), followed by a short period of practice. Extensive training with vOICe confers blind subjects with some ability to distinguish high contrast shapes on a clean background (*Auvray et al., 2007*; *Striem-Amit et al., 2012*). Conceivably an experienced vOICe user might perform better on the tests described here. Other investigators can attempt to demonstrate this using our published code and specifications (*Liu and Meister, 2018*; copy archived at https://github.com/elifesciences-publications/CARA_Public).

## Discussion

'Seeing is knowing what is where by looking' (*Marr, 1982*). The cognitive assistant CARA enables exactly this process while replacing the eye and early visual system. CARA conveys 'what' by the names of objects and 'where' by the location from where each object calls. 'Looking' occurs when the user actively requests these calls. The principal reason sighted people rely on vision much more than audition is that almost all objects in the world emit useful light signals almost all the time, whereas useful sound signals from our surroundings are few and sporadic. CARA can change this calculus fundamentally, such that all the relevant objects emit easily interpretable sounds. It is conceivable that the extended use of such an intuitive modality eventually produces subjective qualia that resemble the feeling of seeing.

The use of spatialized sound in assistive devices has been considered for some time (*Spagnol et al., 2018*), although to date there are few practical implementations with test results. For the purpose of outdoor navigation, *Loomis et al. (2005)* tested a system that guides the user along a path by virtual sounds emitted from way points. Their subjects preferred speech sounds over simple tones, and performed better under those conditions, which supports the use of voices in CARA's virtual guide. To enable recognition of scenes and objects, various schemes have been considered that convert the scene into spatialized sound, but they largely encode low-level geometric features. At the simplest level, the vOICe system takes a photograph, interprets it as a spectrogram, and presents the resulting sound with a left-to-right scan (*Haigh et al., 2013*; *Meijer, 1992*). Another system parses the scene into inclined planes and encodes each plane with a sequence of musical sounds; a chair might consist of two planes (*Ribeiro et al., 2012*). These designs still do not do enough to 'off-load the blind traveler's brain of . . . arduous tasks' (*Collins, 1985*). Instead of posing acoustic puzzles of this kind, CARA uses plain language to tell the user what is out there and where it is. As a rule, assistive devices have been designed to perform one well-circumscribed

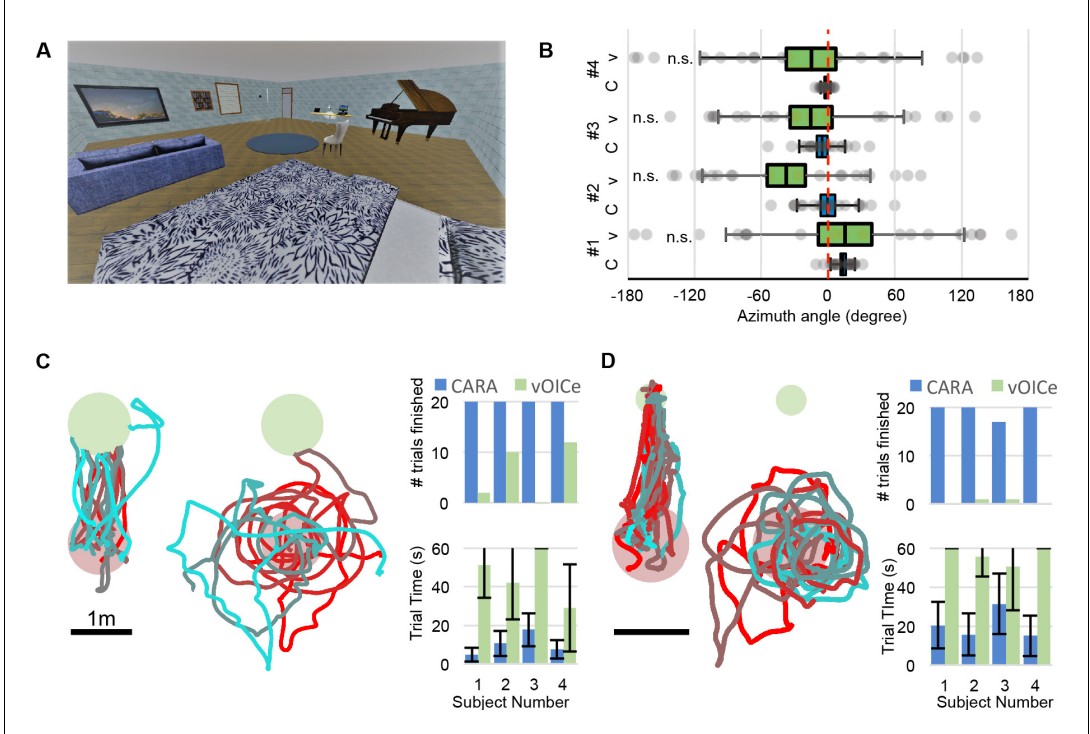

**Figure 5.** Benchmark testing environment. (**A**) A virtual living room including 16 pieces of furniture and other objects. (**B**) Localization of a randomly chosen object relative to the true object location (0 deg, dashed line) for four subjects using CARA (**C**) or vOICe (**V**). Box denotes s.e.m., whiskers s.d. For all subjects the locations obtained with vOICe are consistent with a uniform circular distribution (Rayleigh z test, p>0.05). (**C**) Navigation toward a randomly placed chair. Trajectories from one subject using CARA (left) and vOICe (middle), displayed as in *Figure 3C*. Right: Number of trials completed and time per trial (mean ±s.d.). (**D**) Navigation toward a randomly placed key on the floor (small green circle). Trajectories and trial statistics displayed as in panel C.

DOI: https://doi.org/10.7554/eLife.37841.015

The following figure supplement is available for figure 5:

**Figure supplement 1.** Benchmark tests in a virtual environment.

DOI: https://doi.org/10.7554/eLife.37841.016

function, such as obstacle avoidance, or route finding, or object recognition, or reading of signage (*Loomis et al., 2012*; *Roentgen et al., 2008*). Our main contribution here is to show that augmented reality with object voices offers a natural and effortless human interface for all these functionalities, implemented in a single device.

So far, we have focused on indoor applications. Blind people report that outdoor navigation is supported by many services (access vans, GPS, mobile phones with navigation apps) but these all fall away when one enters a building (*Karimi, 2015*). In its present form CARA can already function in this underserved domain, for example as a guide in a large public building, hotel, or mall. No physical modifications of the space are required. The virtual guide can be programmed to offer navigation options according to the known building geometry. Thanks to the intuitive interface, naive visitors could pick up a device at the building entrance and begin using it in minutes. In this context, recall that our subjects were chosen without prescreening, including cases of early and late blindness and various hearing deficits (*Figure 1D*): They represent a small but realistic sample of the expected blind user population.

The functionality of this system can be enhanced far beyond replacing vision, by including information that is not visible. As a full service computer with online access, the HoloLens can be programmed to annotate the scene and offer ready access to other forms of knowledge. Down the line one can envision an intelligent cognitive assistant that is attractive to both blind and sighted users, with somewhat different feature sets. Indeed this may help integrate the blind further into the community. By this point, we expect that the reader already has proposals in mind for enhancing the

cognitive assistant to communicate things that we cannot see. A hardware/software platform is now available to rapidly implement those ideas and test them with human subjects.

Finally, we demonstrated a separate platform for standardized tests to evaluate and benchmark CARA and other assistive technologies on the same tasks. The use of virtual reality on an affordable consumer device can recreate the identical test environment for blind subjects in laboratories around the world, allowing quantitative performance comparisons even with implanted prosthetic devices. We hope that other developers of assistive technology will engage and collaborate to produce an agreed-upon standard test suite, starting with our published example. This in turn should stimulate competition to exploit the ongoing revolution in wearable computers and machine vision toward creative solutions in the domain of assistive technology.

## Materials and methods

### General implementation of CARA
The hardware platform for the cognitive assistant is the Microsoft HoloLens Development Edition, without any modifications. This is a self-contained wearable augmented reality (AR) device that can map and store the 3D mesh of an indoor space, localize itself in real time, and provide spatialized audio and visual display (*Hoffman, 2016*). We built custom software in Unity 2017.1.0f3 (64-bit) with HoloToolkit-Unity-v1.5.5.0. The scripts are written in C# with MonoDevelop provided by Unity. The experiments are programmed on a desktop computer running Windows 10 Education and then deployed to Microsoft HoloLens. The software is versatile enough to be easily deployed to other hardware platforms, such as AR enabled smart phones.

### User interface
Before an experiment, the relevant building areas are scanned by the experimenter wearing the HoloLens, so the system has a 3D model of the space ahead of time. For each object in the scene, the system creates a voice that appears to emanate form the object's location, with a pitch that increases inversely with object distance. Natural spatialized sound is computed based on a generic head-related transfer function (*Wenzel et al., 1993*); nothing about the software was customized to individual users. Object names and guide commands are translated into English using the text-to-speech engine from HoloToolkit. The user provides input by moving the head to point at objects, pressing a wireless Clicker, using hand gesture commands or English voice commands.

In addition to instructions shown in the main body of the article, non-spatialized instructions are available at the user's request by voice commands. The user can use two voice commands (e.g. 'direction', 'distance') to get the direction of the current object of interest or its distance. Depending on the mode, the target object can be the object label of user's choice (Target Mode) or the virtual guide. 'Turn-by-turn' instructions can be activated by voice commands (e.g. 'instruction'). The instruction generally consists of two parts, the distance the user has to travel until reaching the current target waypoint, and the turn needed to orient to the next waypoint (*Figure 3—figure supplement 2E*).

### Experimental design
All results in *Figures 1–4* were gathered using a frozen experimental protocol, finalized before recruitment of the subjects. The tasks were fully automated, with dynamic instructions from the HoloLens, so that no experimenter involvement was needed during the task. Furthermore, we report performance of all subjects on all trials gathered this way. Some incidental observations and anecdotes from subject interviews are provided in Supplementary Observations. All procedures involving human subjects were reviewed and approved by the Institutional Review Board at Caltech. All subjects gave their informed consent to the experiments, and where applicable to publication of videos that accompany this article.

### Measurement
Timestamps are generated by the internal clock of the HoloLens. The six parameters of the subject's head location and orientation are recorded at 5 Hz from the onset to the completion of each trial in each task. All performance measures are derived from these time series. Localization errors of the

HoloLens amount to <4 cm (*Liu et al., 2018*), which is insignificant compared to the distance measures reported in our study, and smaller than the line width in the graphs of trajectories in *Figures 3* and *4*.

## Task design

*Task 1, object localization (Figure 1):* In each trial, a single target is placed 1 m from the subject at a random azimuth angle drawn from a uniform distribution between 0 and 360 degrees. To localize the target, the subject presses the Clicker to hear a spatialized call from the target. After aiming the face at the object the subject confirms via a voice command ('Target confirmed'). When the location is successfully registered, the device plays a feedback message confirming the voice command and providing the aiming error. The subject was given 10–15 practice trials to learn the interaction with CARA, followed by 21 experimental trials. To estimate the upper limit on performance in this task, two sighted subjects performed the task with eyes open: this produced a standard deviation across trials of 0.31 and 0.36 degrees, and a bias of 0.02 and 0.06 degrees. That includes instrumentation errors as well as uncertainties in the subject's head movement. Note that these error sources are insignificant compared to the accuracy and bias reported in *Figures 1* and *2*.

*Task 2, spatial memory (Figure 2):* This task consists of an exploration phase in which the subject scans the scene, followed by a recall phase with queries about the scene. Five objects are placed two meters from the subject at azimuth angles of −60°, −30°, 0°, 30°, 60° from the subject's initial orientation. Throughout the experiment, a range between −7.5° and 7.5° in azimuth angle is marked by 'sonar beeps' to provide the subject a reference orientation. During the 60 s exploration phase, the subject uses 'Spotlight Mode': This projects a virtual spotlight cone of 30° aperture around the direction the subject is facing and activates object voices inside this spotlight. Typically subjects scan the virtual scene repeatedly, while listening to the voices. In the recall phase, 'Spotlight Mode' is turned off and the subject performs four recall trials. For each recall trial, the subject presses the Clicker, then a voice instruction specifies which object to turn to, the subject faces in the recalled direction, and confirms with a voice command ('Target confirmed'). The entire task was repeated in two blocks that differed in the arrangement of the objects. The object sequence from left to right was 'piano', 'table', 'chair', 'lamp', 'trash bin' (block 1), and 'trash bin', 'piano', 'table', 'chair',' lamp' (block 2). The center object is never selected as a recall target because 0° is marked by sonar beeps and thus can be aimed at trivially.

*Task 3, direct navigation (Figure 3):* In each trial, a single chair is placed at 2 m from the center of the arena at an azimuth angle randomly drawn from four possible choices: 0°, 90°, 180°, 270°. To start a trial, the subject must be in a starting zone of 1 m diameter in the center. During navigation, the subject can repeatedly press the Clicker to receive a spatialized call from the target. The trial completes when the subject arrives within 0.5 m of the center of the target. Then the system guides the subject back to the starting zone using spatialized calls emanating from the center of the arena, and the next trial begins. Subjects performed 19–21 trials. All blind subjects moved freely without cane or guide dog during this task.

To measure performance on a comparable search without CARA, each subject performed a single trial with audio feedback turned off. A real chair is placed at one of the locations previously used for virtual chairs. The subject wears the HoloLens for tracking and uses a cane or other walking aid as desired. The trial completes when the subject touches the target chair with a hand. All blind subjects used a cane during this silent trial.

*Task 4, long range guided navigation (Figure 4):* The experimenter defined a guide path of ~36 m length from the first-floor lobby to the second-floor office by placing nine waypoints in the pre-scanned environment. In each trial, the subject begins in a starting zone within 1.2 m of the first waypoint, and presses the Clicker to start. A virtual guide then follows the trajectory and guides the subject from the start to the destination. The guide calls out 'follow me' with spatialized sound every 2 s, and it only proceeds along the path when the subject is less than 1 m away. Just before waypoints 2–8, a voice instruction is played to inform the subject about the direction of turn as well as approaching stairs. The trial completes when the subject arrives within 1.2 meters of the target. Voice feedback ('You have arrived') is played to inform the subject about arrival. In this task all blind subjects used a cane.

*Free walking:* To measure the free walking speed, we asked subjects to walk for 20 m in a straight line in an unobstructed hallway using their preferred walking aid. Subjects 1 and 2 used a guide dog, the others a cane.

## Data analysis and visualization

MatLab 2017b (Mathworks) and Excel (Microsoft) were used for data analysis and visualization. Unity 5.6.1f1 was used to generate 3D cartoons of experiments and to visualize 3D trajectories. Photoshop CC 2017 was used for overlaying trajectories on floor plans.

*Aiming:* In tasks 1 and 2, aiming errors are defined as the difference between the target azimuth angle and the subject's front-facing azimuth angle. In task 2, to correct for the delay of voice command registration, errors are measured at 1 s before the end of each trial.

*Trajectory smoothing:* The HoloLens tracks its wearer's head movement, which includes lateral movements perpendicular to the direction of walking. To estimate the center of mass trajectory of the subject we applied a moving average with 2 s sliding window to the original trajectory.

*Length of trajectory and deviation index:* In the directed navigation task and the long-range guided navigation task we computed the excess distance traveled by the subject relative to an optimal trajectory or the guide path. The deviation index, $DI$, is defined as

$$DI = \frac{L_{exp} - L_{ref}}{L_{ref}}$$

where $L_{exp}$ is the length of the trajectory measured by experiment and $L_{ref}$ is the length of the reference trajectory. A value near 0 indicates that the subject followed the reference trajectory well.

In the direct navigation task, we divided each trial into an orientation phase where the subject turns the body to face the target, and a navigation phase where the subject approaches the target. We calculated head orientation and 2D distance to target in each frame, and marked the onset of the navigation phase when the subject's distance to target changed by 0.3 m. Note that with this criterion the navigation phase includes the occasional trajectory where the subject starts to walk in the wrong direction. In this task, $L_{ref}$ is defined as the length of the straight line from the subject's position at the onset of the navigation phase to the nearest point of the target trigger zone.

In the long-range guided navigation task, $L_{ref}$ is the length of the guide trajectory. Due to variability in placing waypoints and tracking, the length of guide trajectories varied slightly across subjects ($L_{ref}$=36.4 ± 0.7 m, mean ±s.d.). Negative $DI$ values are possible in this task if the subject cuts corners of the guide trajectory.

*Speed:* Speed is calculated frame-by-frame using the displacements in the filtered trajectories. For the long-range guided navigation task, which includes vertical movements through space, the speed of translation is computed in three dimensions, whereas for the other tasks that occur on a horizontal plane we did not include the vertical dimension. For all tasks, we estimated walking speed by the 90th percentile of the speed distribution, which robustly rejects the phases where the subject chooses an orientation. The normalized speed is obtained by dividing this value by the free walking speed.

## Supplementary observations

Here, we report incidental observations during experiments with CARA that were not planned in the frozen protocol, and comments gathered from blind subjects in the course of the experiments.

*Subject 1:* During navigation with the virtual guide says "seems to me the 'follow me' sound means keep going straight". Thinks addition of GPS services could make the system useful outdoors as well. Suggests experimenting with bone conduction headphones. Offers us 1 hr on his radio show.

*Subject 2:* During direct navigation says "pitch change [with distance] was informative". During navigation with the virtual guide says "'Follow me' was too much information". Prefers to follow the explicit turn instructions. She could then transmit those instructions to her guide dog.

*Subject 3:* In addition to object voices, he likes instructions of the type 'keep going forward for xx meters'. During a previous visit using a similar system he commented on possible adoption by the blind community: "I could see people spending in 4 figures for [something] light and reliable, and use it all the time". Also supports the concept of borrowing a device when visiting a public building

or mall. Devices in the form of glasses would be better, preferably light and thin. "Use the computing power of my phone, then I don't have to carry anything else." Likes the external speakers because they don't interfere with outside sound. Finds it easy to localize the virtual sound sources.

*Subject 4:* After navigation with the virtual guide says "That was fun. When can I get one?" Primarily used the 'follow me' voice, and the cane to correct for small errors. Reports that the turn instructions could be timed earlier (this is evident also in Video 1). On a previous visit using a similar system: "I'm very excited about all of this, and I would definitely like to be kept in the loop". Also suggests the system could be used in gaming for the blind.

*Subject 5:* During navigation with the virtual guide realized she made a wrong turn (see Fig. 4C) but the voice made her aware and allowed her to correct. Reports that the timing of turn instructions is a little off.

*Subject 6:* After all tasks says "That was pretty cool" and "The technology is there."

*Subject 7:* On the second trial with the virtual guide reports that she paid more attention to the 'follow me' sound (she strayed temporarily on the first trial, Fig. 4B). Wonders whether the object voices will be strong enough in a loud environment.

## Benchmarking platform using virtual reality

The benchmarking platform runs on the HTC Vive VR headset and a Windows 10 desktop computer. A TPCast wireless adapter (CE-01H, https://www.tpcastvr.com/) replaces the standard headset cable so the subject can move freely within the 4 m x 4 m square arena. An Xbox One S controller is connected wirelessly to the host computer for the subject to start trials, confirm aiming, and control the modes of CARA. Audio is delivered by a pair of wireless headphones (SONY WH-1000XM2). All code and data to replicate this environment and the reported tests is publically available at https://github.com/meisterlabcaltech/CARA_Public (*Liu and Meister, 2018*; copy archived at https://github.com/elifesciences-publications/CARA_Public).

All three benchmarking tasks are set in a 10 m x 10 m virtual environment that simulates a living room with 16 objects labeled with sound tags. The rendered scenes are close to photo-realistic, and computer vision algorithms trained on real scenes will correctly identify objects in the virtual scene as well (*Figure 3—figure supplement 2F*). The first benchmark task (*Figure 5B*) resembles Task 1 (see section on Task Design above) except that the aiming target is chosen at random from the 16 objects in the studio. CARA users have access to Spotlight and Target Mode. The second benchmark task (*Figure 5C*) resembles Task 3 with a 60 s time limit on each trial. CARA users only have Target Mode available for this task. To facilitate detection with vOICe the chair was made white, and left unoccluded by any other objects. The third benchmark task (*Figure 5D*) replaces the chair in the second benchmark task with a key on the floor. Within CARA, Target Mode and Spotlight Mode can be used in this task. Two conditions have to be met to finish a trial: (1) the head of the subject is within 1 m of the key and (2) the subject faces within 30 degrees of the key. To accomplish this the subject must bend down or kneel facing the key. At this point, a simple reach with the hand would allow grasping a real object. A trial fails if not finished within 60 s. To avoid excess frustration among the subjects a task is terminated after five failed trials.

### Voice control on CARA

In addition to the Clicker, subjects can also use natural language (e.g. English) as input to the system. Two subsystems of voice input are implemented: 1) keyword recognition (PhraseRecognitionSystem) monitors in the background what the user says, detects phrases that match the registered keywords, and activates corresponding functions on detection of keyword matches. 2) dictation (DictationRecognizer) records what the user says and converts it into text. The first component enables subjects to confirm their aiming in the object localization task and mental imagery task with the voice command "target confirmed". It also enables the experimenter to control the experiment at runtime.

Keywords and their functions are defined through adding keywords to the keyword manager script provided by HoloToolkit and editing their responses. The KeywordRecognizer component starts at the beginning of each instance of the application and runs in the background throughout the instance of the application except for the time period in which dictation is in use.

To allow users to create object labels, the DictationRecognizer provided by HoloToolkit is used to convert natural language spoken by the user to English text. Due to the mutual exclusivity, KeywordRecognizer is shut down before DictationRecognizer is activated, and restarted after the dictation is finished.

### Automated wayfinding

In addition to hand-crafting paths, we implemented automated wayfinding by taking advantage of Unity's runtime NavMesh 'baking' which calculates navigable areas given a 3D model of the space. At runtime, we import and update the 3D mesh of the scanned physical space and use it to bake the 3D mesh. When the user requests guided navigation, a path from the user's current location to the destination of choice is calculated. If the calculated path is valid, the virtual guide guides the user to the destination using the computer-generated path.

### Cost of the CARA system

The hardware platform used in the research – Microsoft HoloLens Development Edition – currently costs $3000. Several comparable AR goggles are in development, and one expects their price to drop in the near future. In addition, smart phones are increasingly designed with AR capabilities, although they do not yet match the HoloLens in the ability to scan the surrounding space and localize within it.

### Battery and weight

The current HoloLens weighs 579 g. Like all electronic devices, this will be further miniaturized in the future. The current battery supports our system functions for 2–5 hr, sufficient for the indoor excursions we envision in public buildings, led by the 'virtual guide'. A portable battery pack can extend use to longer uninterrupted sessions.

### Tracking robustness

While in most indoor scenarios that we have tested the tracking of HoloLens was reliable and precise, we have encountered occasional loss of tracking or localization errors. This occurs particularly when the environment lacks visual features such as a narrow space with white walls.

### Extensions

Because this cognitive assistant is largely defined by software its functionalities are very flexible. For example, the diverse recommendations from subjects noted above (Supplementary Observations) can be implemented in short order. In addition, one can envision hardware extensions by adding peripherals to the computer. For example, a haptic belt or vest could be used to convey collision alarms (*Adebiyi et al., 2017*), thus leaving the auditory channel open for the highly informative messages.

## Data and materials availability

Data and code that produced the figures are available on the Dryad Digital Repository.

## Acknowledgements

Supported by grant 103212 from the Shurl and Kay Curci Foundation. We thank Ralph Adolphs, David Anderson, and Shin Shimojo for comments on the manuscript, and Kristina Dylla for posing for *Figure 1A* and field testing all of the tasks.

## Additional information

### Competing interests

Yang Liu, Markus Meister: is an author on a patent application (Patent Application No. 15/852,034) 'Systems and Methods for Generating Spatial Sound Information Relevant to Real-World Environments'. The other author declares that no competing interests exist.

## Funding

| Funder | Grant reference number | Author |
|---|---|---|
| Shurl and Kay Curci Foundation | 103212 | Markus Meister |

The funders had no role in study design, data collection and interpretation, or the decision to submit the work for publication.

## Author contributions

Yang Liu, Conceptualization, Data curation, Software, Formal analysis, Validation, Investigation, Visualization, Methodology, Writing—review and editing; Noelle RB Stiles, Resources, Validation, Investigation, Methodology; Markus Meister, Conceptualization, Supervision, Funding acquisition, Writing—original draft, Project administration, Writing—review and editing

## Author ORCIDs

Yang Liu http://orcid.org/0000-0002-8155-9134
Markus Meister http://orcid.org/0000-0003-2136-6506

## Ethics

Human subjects: All procedures involving human subjects were reviewed and approved by the Institutional Review Board at Caltech, Human Subjects Protocol 16-0663. All subjects gave their informed consent to the experiments, and where applicable to publication of videos that accompany this article.

## Decision letter and Author response

Decision letter https://doi.org/10.7554/eLife.37841.021
Author response https://doi.org/10.7554/eLife.37841.022

# Additional files

## Supplementary files

• Transparent reporting form
DOI: https://doi.org/10.7554/eLife.37841.017

## Data availability

All data generated or analysed during this study are included in the manuscript and supporting files. Source data files for all figures have been deposited in Dryad. Additional code is published on a Github repository (https://github.com/meisterlabcaltech/CARA_Public; copy archived at https://github.com/elifesciences-publications/CARA_Public).

The following dataset was generated:

| Author(s) | Year | Dataset title | Dataset URL | Database and Identifier |
|---|---|---|---|---|
| Liu Y, Stiles NRB, Meister M | 2018 | Data from: Augmented Reality Powers a Cognitive Prosthesis for the Blind | https://dx.doi.org/10.5061/dryad.8mb5r88 | Dryad Digital Repository, 10.5061/dryad.8mb5r88 |

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
