## [Decision Letter]

[Editors’ note: a previous version of this study was rejected after peer review, but the authors submitted for reconsideration. The first decision letter after peer review is shown below.]

Thank you for submitting your work entitled "Augmented Reality Powers a Cognitive Prosthesis for the Blind" for consideration by *eLife*. Your article has been reviewed by three peer reviewers, one of whom is a member of our Board of Reviewing Editors, and the evaluation has been overseen by a Senior Editor.

Our decision has been reached after consultation between the reviewers. Based on these discussions and the individual reviews below, we regret to inform you that your work will not be considered further for publication in *eLife*.

Two main concerns emerged in the reviews and consultation among reviewers. The first regards overlap of the proposed system with other similar approaches. This overlap is not discussed sufficiently in the paper to evaluate how much of an advance the current system offers. The second is that the paper does not introduce a set of benchmarks that could be used to compare different systems and describing the limitations of the current system in more detail.

*Reviewer #1:*

This paper presents an interesting approach to help restore independence to people with significant visual impairments. The approach centers around proving auditory guidance about the location of objects in an environment, or about a desired path through an environment. The work appears well done and the paper is clearly written. My main concerns have to do with novelty.

The Discussion states that the main contribution of this paper is to show that devices such as the one proposed can provide a substrate for many visual tasks, while previous work has tended to focus on specific tasks. I am not an expert in this area, but a cursory look through the literature brings up several papers using quite similar technology (e.g. using the Kinect system and auditory feedback for avoiding obstacles – Saputra et al., 2014). I think the impact of the present work would be enhanced considerably if more visual tasks were probed and the abilities and limitations of this approach were laid out in more detail. This would not necessarily require visually impaired subjects. Examples might include identifying objects in cluttered environments, navigating a cluttered environment, reaching or grasping an object. In many ways, the tasks probed in the paper were too easy for the subjects and do not define the capabilities of the approach. Are there a few tasks that could be standardized and then used to compare different approaches? Several of the tasks in the paper get close to this, but both a larger complement of tasks and clearer standardization of task conditions might serve to set a clearer performance standard.

*Reviewer #2:*

The study describes a system for sensory substitution: auditory guidance for the blind. It is erroneously called prosthesis, even though instead of restoring sight it just replaces it with the auditory guidance. The term "prosthesis" should be replaced with "auditory substitute" or something along these lines, throughout the manuscript.

"Here we present such an approach that bypasses the need to convey the sensory data entirely, and focuses instead on the important high-level knowledge, presented at a comfortable data rate and in an intuitive format."

Sensory substitution and augmentation certainly has its utility, but is unlikely to replace the richness of the visual input (if it could be restored).

The manuscript does not mention a related system already on the market, called Orcam. It has a subset of features mentioned in this manuscript, and others could be added.

The manuscript also does not cite a previous study about auditory guidance for the blind – "Assessment of feedback modalities for wearable visual aids in blind mobility", by A. Adebiyi et al., 2017. Nor does it mention the whole set of studies about auditory encoding of the space from A. Amedi's lab: http://brain.huji.ac.il/publications.asp

*Reviewer #3:*

The study by Meister and colleagues proposes a novel approach to a visual prosthesis, by substituting objects in the environment with spatially localized voices. They demonstrate the utility of this in a set of simplified scenarios, showing that the user can identify, localize, and navigate based on these localized voice cues. The study does not address the (probably more challenging) issue of converting a visual scene into voice cues.

This is a clever idea and is likely to have a significant impact, particularly because this is a technology that can be implemented imminently, safely, and relatively cheaply, as opposed to approaches such as retinal prostheses or gene therapies. My comments are centered around clarifying exactly what has been achieved in the current study. These are relatively minor comments for what is overall an intriguing study.

Primarily this centers around the claim in the Abstract which states "A wearable computer captures video and other data, extracts the important scene knowledge, and conveys that.…" Yet the experiments presented do not use the video to generate the cues, they are either virtual objects or a precomputed path through a building. Only in the Discussion (Technical extensions) do they note that the first half of the problem has not yet been solved. This should be corrected in the abstract and noted explicitly early on, as well as the fact that tests were performed with virtual objects. It is also not clear at many points in the text that "targets are placed" means virtual, rather than physical targets.

Likewise, the prosthesis is described as conveying information at the cognitive level, however most of what they present is better described as a hybrid of sensory substitution with cognitive labels, as it is largely dependent on auditory spatial localization. Indeed, their first experiment is simply confirming that a subject can point towards a synthesized auditory stimulus. The only aspects where the language component really comes into play is the spatial memory of 5 objects, or warning signals for navigation. It would be helpful to explicitly point out the key role auditory spatial localization plays.

Finally, a major challenge that should be at least discussed (or even better, tested) is that real scenes do not have just 1-5 isolated objects, but actually many more. I can imagine ways that this could be addressed, but it would be helpful to be explicit about this and other challenges in moving from simple virtual environments to the real world.

[Editors’ note: what now follows is the decision letter after the authors submitted for further consideration.]

Thank you for submitting your article "Augmented Reality Powers a Cognitive Prosthesis for the Blind" for consideration by *eLife*. Your article has been reviewed by three peer reviewers, including Fred Rieke as the Reviewing Editor, and the evaluation has been overseen by Eve Marder as the Senior Editor. The following individual involved in review of your submission has agreed to reveal their identity: Cristopher M Niell.

The reviewers have discussed the reviews with one another and the Reviewing Editor has drafted this decision to help you prepare a revised submission.

This is a revision of a paper investigating the capabilities of a 'cognitive prosthetic' to help the blind explore their environment. All of the reviewers agreed that the paper had improved considerably. Specifically, the authors have done a good job of answering all of the comments from the previous reviews, and in particular being clear about what their system achieves and its relationship with previous work. The new benchmarking system, and inclusion of a cluttered scene, provides a more stringent test of the device's capabilities and substantially enhances the paper. The reviewers noted a few areas in which the paper could be further improved:

Description of the device: The authors addressed concerns with review of the prior articles, but their description of the system as a "prosthesis" is still a concern.

1) In their response letter, the authors state: "Our device can restore the main function of sight ("knowing what is where by looking") although it does not emulate what sight used to feel like. Similarly, a prosthetic leg restores the function of walking without emulating what the leg used to feel like."

Functionally, prosthetic replacement of a leg is much more similar to the original than vision being replaced with auditory explanation of the scene. Replacement of the missing legs with a wheelchair, which provides mobility but through a different mechanism, is a better analogy. And again, nobody calls a wheelchair a prosthesis.

2) Regarding the title: "cognitive prosthesis" sounds very odd for such a device. For example, many websites and e-book readers provide an audio option, which certainly enables blind people to read, but nobody calls such a reader a "cognitive prosthesis". We suggest replacing "prosthesis" with "assistant", as defined in the name CARA: "cognitive augmented reality assistant for the blind".

3) "Visual prosthesis" is mentioned again at the end of the Abstract, again please replace with a term that is more in keeping with what you have actually done.

Benchmarking: Performance on the benchmarking tasks in Figure 5 with CARA is very good. Did you in any cases increase the task difficulty to see where performance degraded? One specific task that might be interesting is gradually increasing the number of objects in the environment to see at what point and how performance degrades. More broadly, some discussion of what is challenging with the device from any of the users would be welcome.

---

## [Author Response]

[Editors’ note: the author responses to the first round of peer review follow.]

Two main concerns emerged in the reviews and consultation among reviewers. The first regards overlap of the proposed system with other similar approaches. This overlap is not discussed sufficiently in the paper to evaluate how much of an advance the current system offers.

The revised manuscript offers more on this. Related approaches are discussed in the Introduction. Incorporation of other technical developments in the subsection “Technical extensions”. Sensory substitution by vOICe in the subsection “An open-source benchmarking environment for assistive devices”. Other uses of spatialized sound in the Discussion.

The second is that the paper does not introduce a set of benchmarks that could be used to compare different systems and describing the limitations of the current system in more detail.

We have added a new section that introduces a system for benchmarking, allowing researchers anywhere to compare different devices on the same tasks (subsections “An open-source benchmarking environment for assistive devices” and “Benchmarking platform using virtual reality”). We illustrate its use with a side-by-side comparison of two very different approaches to a visual prosthesis, including new experiments on human subjects (see new Figure 5 and Figure 5—figure supplement 1). We also published on GitHub all the specs and code needed to implement the benchmarking system.

Reviewer #1:This paper presents an interesting approach to help restore independence to people with significant visual impairments. The approach centers around proving auditory guidance about the location of objects in an environment, or about a desired path through an environment. The work appears well done and the paper is clearly written. My main concerns have to do with novelty.The Discussion states that the main contribution of this paper is to show that devices such as the one proposed can provide a substrate for many visual tasks, while previous work has tended to focus on specific tasks. I am not an expert in this area, but a cursory look through the literature brings up several papers using quite similar technology (e.g. using the Kinect system and auditory feedback for avoiding obstacles – Saputra et al., 2014).

This example reinforces our claim regarding the integration of many visual functions. That article (with zero citations in Web of Science) only addresses obstacle avoidance. No localization. No scene representation. No guidance. No long-range navigation.

I think the impact of the present work would be enhanced considerably if more visual tasks were probed and the abilities and limitations of this approach were laid out in more detail. This would not necessarily require visually impaired subjects. Examples might include identifying objects in cluttered environments, navigating a cluttered environment, reaching or grasping an object.

Following this suggestion, we built a benchmarking environment and illustrated it with performance tests that cover exactly these three cases (subsections “An open-source benchmarking environment for assistive devices” and “Benchmarking platform using virtual reality”, Figure 5 and Figure 5—figure supplement 1). We performed these tests with two different assistive devices using sighted but blindfolded subjects.

In many ways, the tasks probed in the paper were too easy for the subjects and do not define the capabilities of the approach.

Our blind subjects would not agree; they were surprised how well they did given the challenges, such as crossing a large open space and climbing stairs.

Are there a few tasks that could be standardized and then used to compare different approaches? Several of the tasks in the paper get close to this, but both a larger complement of tasks and clearer standardization of task conditions might serve to set a clearer performance standard.

This suggestion has been implemented in the new section on benchmarking (subsections “An open-source benchmarking environment for assistive devices” and “Benchmarking platform using virtual reality”, Figure 5 and Figure 5—figure supplement 1).

Reviewer #2:The study describes a system for sensory substitution: auditory guidance for the blind. It is erroneously called prosthesis, even though instead of restoring sight it just replaces it with the auditory guidance.

Our device can restore the main function of sight (“knowing what is where by looking”) although it does not emulate what sight used to feel like. Similarly a prosthetic leg restores the function of walking without emulating what the leg used to feel like.

The term "prosthesis" should be replaced with "auditory substitute" or something along these lines, throughout the manuscript.

We have revised the text to use the term “cognitive prosthesis”. This appears in the Oxford English Dictionary as “an electronic computational device that extends the capability of human cognition or sense perception”, exactly what our system aims for. The term “visual prosthesis” appears once when citing Collins (1985) who used it liberally for systems of related design.

"Here we present such an approach that bypasses the need to convey the sensory data entirely, and focuses instead on the important high-level knowledge, presented at a comfortable data rate and in an intuitive format."Sensory substitution and augmentation certainly has its utility, but is unlikely to replace the richness of the visual input (if it could be restored).

It is conceivable that the extended use of such a cognitive prosthesis for a broad range of functions eventually produces a new type of subjective qualia that resemble the feeling of seeing. The Discussion mentions this briefly.

The manuscript does not mention a related system already on the market, called Orcam. It has a subset of features mentioned in this manuscript, and others could be added.

Orcam is another single-feature gadget: a camera attached to an eye frame that can read text and money out loud if you point it in the right direction. No localization. No scene analysis. No obstacle detection. No navigation. There are dozens more of these apps and gadgets, and we cite reviews that cover them (Bujacz and Strumillo, 2016; Jafri et al., 2014; Karimi, 2015; Roentgen et al., 2008).

The manuscript also does not cite a previous study about auditory guidance for the blind – "Assessment of feedback modalities for wearable visual aids in blind mobility", by A. Adebiyi et al., 2017.

Actually this was the first article on our list of references.

Nor does it mention the whole set of studies about auditory encoding of the space from A. Amedi's lab: http://brain.huji.ac.il/publications.asp

In fact we did cite Striem-Amit et al., 2012, which in turn reviews a good number of other papers from Amedi’s group. We have now added another citation to Maidenbaum, Abboud and Amedi, 2014.

Reviewer #3:[…]Primarily this centers around the claim in the Abstract which states "A wearable computer captures video and other data, extracts the important scene knowledge, and conveys that.…" Yet the experiments presented do not use the video to generate the cues, they are either virtual objects or a precomputed path through a building. Only in the Discussion (Technical extensions) do they note that the first half of the problem has not yet been solved. This should be corrected in the abstract and noted explicitly early on, as well as the fact that tests were performed with virtual objects. It is also not clear at many points in the text that "targets are placed" means virtual, rather than physical targets.

The revised manuscript clarifies this and also points to tantalizing recent developments towards real-time object identification in video feed (see Figure 3—figure supplement 2F). We expect that this will lead to a complete front end solution for object acquisition within a year. Meanwhile we want to publish our solution for the back end (communication with the user) because that can be combined with any number of front end solutions. By publishing the human interface now, and drawing attention to the powerful platform of the HoloLens, we enable others whose talents and expertise are on the front end.

Likewise, the prosthesis is described as conveying information at the cognitive level, however most of what they present is better described as a hybrid of sensory substitution with cognitive labels, as it is largely dependent on auditory spatial localization. Indeed, their first experiment is simply confirming that a subject can point towards a synthesized auditory stimulus. The only aspects where the language component really comes into play is the spatial memory of 5 objects, or warning signals for navigation. It would be helpful to explicitly point out the key role auditory spatial localization plays.

Yes, the reason subjects adopt this system so easily is people’s effortless ability to localize sounds combined with the high cognitive content of language labels.

Finally, a major challenge that should be at least discussed (or, even better, tested) is that real scenes do not have just 1-5 isolated objects, but actually many more. I can imagine ways that this could be addressed, but it would be helpful to be explicit about this and other challenges in moving from simple virtual environments to the real world.

The new manuscript section on standardized benchmarking includes tests in a room with many more objects (see Figure 5A). These tests also make more obvious use of the language labels. Within the benchmarking system that we published one can design environments of arbitrary visual complexity.

[Editors’ note: the author responses to the re-review follow.]

The reviewers noted a few areas in which the paper could be further improved:Description of the device: The authors addressed concerns with review of the prior articles, but their description of the system as a "prosthesis" is still a concern.1) In their response letter, the authors state: "Our device can restore the main function of sight ("knowing what is where by looking") although it does not emulate what sight used to feel like. Similarly, a prosthetic leg restores the function of walking without emulating what the leg used to feel like."Functionally, prosthetic replacement of a leg is much more similar to the original than vision being replaced with auditory explanation of the scene. Replacement of the missing legs with a wheelchair, which provides mobility but through a different mechanism, is a better analogy. And again, nobody calls a wheelchair a prosthesis.2) Regarding the title: "cognitive prosthesis" sounds very odd for such a device. For example, many websites and e-book readers provide an audio option, which certainly enables blind people to read, but nobody calls such a reader a "cognitive prosthesis". We suggest replacing "prosthesis" with "assistant", as defined in the name CARA: "cognitive augmented reality assistant for the blind".3) "Visual prosthesis" is mentioned again at the end of the Abstract, again please replace with a term that is more in keeping with what you have actually done.

We have replaced “prosthesis” with “assistant” throughout, except where we refer to an invasive device or where we quote another author.

Benchmarking: Performance on the benchmarking tasks in Figure 5 with CARA is very good. Did you in any cases increase the task difficulty to see where performance degraded? One specific task that might be interesting is gradually increasing the number of objects in the environment to see at what point and how performance degrades.

We have not tried to push the limits of clutter in the environment. This would become more informative once there is a competing technology with performance similar to CARA. In the current version, “spotlight mode” activates objects inside a 30 deg cone. If the user wanted to search the space more finely that parameter could be adjusted; of course this could also be done dynamically during the search.

More broadly, some discussion of what is challenging with the device from any of the users would be welcome.

A supplementary file contains comments from our users regarding perceived challenges and desired features. The revised manuscript has an explicit pointer to this supplement.